# Epigenetic Regulation of ALS and CMT: A Lesson from *Drosophila* Models

**DOI:** 10.3390/ijms22020491

**Published:** 2021-01-06

**Authors:** Masamitsu Yamaguchi, Kentaro Omori, Satoshi Asada, Hideki Yoshida

**Affiliations:** 1Department of Applied Biology, Kyoto Institute of Technology, Matsugasaki, Sakyo-ku, Kyoto 606-8585, Japan; omori.kentaro.83r@st.kyoto-u.ac.jp (K.O.); msc2013722@gmail.com (S.A.); 2Kansai Gakken Laboratory, Kankyo Eisei Yakuhin Co. Ltd., Seika-cho, Kyoto 619-0237, Japan

**Keywords:** amyotrophic lateral sclerosis, Charcot–Marie–Tooth disease, epigenetics, *Drosophila melanogaster*, genetic screening, long noncoding RNA, histone modification enzyme

## Abstract

Amyotrophic lateral sclerosis (ALS) is the third most common neurodegenerative disorder and is sometimes associated with frontotemporal dementia. Charcot–Marie–Tooth disease (CMT) is one of the most commonly inherited peripheral neuropathies causing the slow progression of sensory and distal muscle defects. Of note, the severity and progression of CMT symptoms markedly vary. The phenotypic heterogeneity of ALS and CMT suggests the existence of modifiers that determine disease characteristics. Epigenetic regulation of biological functions via gene expression without alterations in the DNA sequence may be an important factor. The methylation of DNA, noncoding RNA, and post-translational modification of histones are the major epigenetic mechanisms. Currently, *Drosophila* is emerging as a useful ALS and CMT model. In this review, we summarize recent studies linking ALS and CMT to epigenetic regulation with a strong emphasis on approaches using *Drosophila* models.

## 1. Introduction

Amyotrophic lateral sclerosis (ALS) is a fatal disorder characterized by the degeneration of both upper and lower motor neurons in the spinal cord, brainstem, and motor cortex [1]. Rapid progress of neurodegeneration induces muscle weakness, atrophy, and spasticity that finally result in death within several years after disease onset. ALS is sometimes associated with frontotemporal dementia (FTD), a clinically diverse dementia syndrome accompanied by behavioral changes and language dysfunction. ALS and FTD are considered to form a continuum of broad neurodegenerative disorders [2]. Charcot–Marie–Tooth disease (CMT) is one of the most common hereditary motor and sensory neuropathies. The average prevalence of CMT is approximately 1:2500 persons, while that of ALS is about 1:25,000 persons all over the world [3]. In general, CMT demonstrates early onset in childhood, with relatively slow progression of sensory and distal muscle defects. Of note, the severity and progression of CMT symptoms markedly vary among individuals [4]. Some CMT patients exhibit demyelination (demyelinating type) and others have a reduced number of axons with no evidence of demyelination (axonal-type) [5,6]. Several types of CMT were summarized in previous reviews [3,5,6]. No effective therapy has been established for ALS/FTD and CMT [3,7,8]. The phenotypic heterogeneity of ALS and CMT suggests the presence of mechanisms (modifiers) that alter the characteristics of these diseases such as site and age of onset, progression rate, and duration of disease [3,9]. Identification of these modifiers is important because they may be targets for the development of therapies. Epigenetic regulation of biological functions via gene expression without alterations in the DNA sequence [10] may be one such modifier. Alterations of epigenetic regulation disrupt normal signaling pathways that may serve as important markers of initiation and/or progression in many human diseases, including neuropathies [11,12,13].

*Drosophila* has a relatively short life span and produces many offspring, facilitating statistical analyses of the experimental data. It is well known that nearly 75% of human disease-related genes have functional orthologues in *Drosophila* [14,15]. Many mutants and RNAi lines available from stock centers enable extensive genetic screening. Importantly, there is less ethical concern regarding experiments because insects lie outside of animal laws in many countries. *Drosophila* is thus emerging as a useful model to study human diseases [14,15]. Both ALS and CMT induce neurodegeneration. In ALS, both upper and lower neurons are degenerated, whereas only lower neurons are affected in CMT. However, *Drosophila* models for ALS and CMT established thus far have similar phenotypes, possibly because of the shorter neurons in *Drosophila* than in humans. In this review, we overview human cell and animal models to investigate epigenetic regulation of ALS and CMT, mainly focusing on *Drosophila* models, although studies of epigenetic regulation involved in both diseases are in their initial stages compared with those of other human diseases. In writing this review, we searched the references by PubMed using the keywords described above.

## 2. *Drosophila* ALS Models

Approximately 10% of ALS cases are familial ALS and the other 90% are sporadic [1]. More than 30 ALS-causative genes, such as *chromosome 9 open reading frame 72* (*C9orf72*), *superoxide dismutase 1* (*SOD1*), *TAR DNA-binding protein 43* (*TDP-43*), and *fused in sarcoma* (*FUS*), have been identified [16,17]. These genes play important roles in the pathogenesis of both familial and sporadic ALS.

Various *Drosophila* models of ALS targeting *SOD1* [18], *FUS* [19], *VAMP-associated protein B* (*VAPB*) [20], *TDP-43* [21], *Valosin-containing protein* (*VCP*)/*p97* [22], *Ubiquilin-2* (*UBQLN2*) [23], *C9orf72* [24], and some of their *Drosophila* homologues have been established and analyzed [8]. Of note, many of these *Drosophila* ALS models exhibit common phenotypes such as larval crawling defect, adult climbing defect, aberrant morphology of synapse at the neuromuscular junction (NMJ), and aberrant eye morphology (rough eye phenotype) [8]. Genetic screening using ALS model flies identified several genes that genetically interact with causative genes of ALS. The genes involved in the EGFR signaling pathway, such as *rhomboid-1*, *rhomboid-3*, or *mirror*, genetically interacted with *cabeza* (*caz*), a *Drosophila* homologue of *FUS* [25]. Genetic screening also identified the *wallenda* gene, encoding a conserved MAPKKK, as a modifier of TDP-43 [26]. A genetic link between *caz* and *hippo* (*hpo*), the *Drosophila* orthologue of human *Mammalian sterile 20-like kinase 1* and *2*, has also been reported [27]. In addition to a role in tumor suppression, current studies revealed a novel role of Hippo in synaptic development that may be independent of Yorkie, a downstream target of the canonical Hippo pathway [28]. Aberrant morphology of mitochondria, such as fragmentation, is reported both in ALS patients and *Drosophila* expressing human TDP-43 [29,30]. Genes involved in mitochondrial dynamics, such as *Drosophila Mitofusion* (*dMfn*)*/Marf*, *Opa1*, and *Drp1*, were identified as modifiers of TDP-43 or FUS-induced phenotypes in *Drosophila* [29].

## 3. *Drosophila* CMT Models

Current studies employing next-generation sequencing identified more than 80 genes associating with CMT [31]. As *Drosophila* lacks myelin sheaths and Schwann cells surrounding the axon of neurons, it is only suitable as axonal-type CMT models [3]. More than 35 *Drosophila* homologues of causative genes of CMT in humans have been identified. Many of them may be causative genes specific for the formation of abnormal axonal phenotypes observed in axonal-type CMT [3].

Various *Drosophila* models of CMT targeting different genes have been established and recently reviewed in detail: genes encoding proteins involved in mitochondrial dynamics, such as Mitofusin 2 (MFN2), Ganglioside-induced differentiation associated protein 1 (GDAP1), and solute carrier family 25 member 46 (SLC25A46); genes encoding proteins involved in lysosome and endosome functions such as Factor-induced gene 4 (FIG4) and RAB7; genes encoding aminoacyl-tRNA synthetases; genes encoding proteins involved in axonal transport such as Kinesin family member 1A (KIF1A) [32]. Many of these *Drosophila* CMT models exhibit common phenotypes such as larval crawling defect, adult climbing defect, and aberrant morphology of synapses at the NMJ [3,32]. A genetic link between *Drosophila FIG4* (*dFIG4*) and *hpo* was also reported, suggesting that *hpo* is involved in a common pathway in the pathogenesis of CMT and ALS [33].

## 4. Epigenetic Regulation

Epigenetic regulation can modify biological functions via gene expression without altering the DNA sequence [10]. In the last decade, numerous mechanisms underlying epigenetic processes, such as DNA methylation, post-translational modification of histones, and noncoding RNA, have been extensively investigated [11,34,35]. Epigenetic alterations with consequent disruption of normal signaling pathways serve as important markers for the initiation and/or progression of many human diseases, including neuropathies [11,12,13].

### 4.1. DNA Methylation

DNA methylation refers to the process of adding a methyl group to cytosine to form 5-methylcytosine and is involved in transcriptional regulation [11,12,36]. DNA methylation at promoters of genes generally plays a role in transcriptional silencing, whereas that in the coding region can activate transcription. In addition to these genomic regions, recent genome-wide mapping of DNA methylation revealed that it is associated with other genomic regions such as introns, splicing sites, specific chromatin domains, and other intragenic/intergenic elements [12]. DNA methylation is thus responsible for a wide range of biological processes, including positive and negative regulation of transcription, repression of transposable elements, genomic imprinting, X-chromosome inactivation, and maintenance of genome stability [12].

### 4.2. Noncoding RNAs

Micro RNAs (miRNAs), noncoding short pieces of RNA, generally bind to the 3′-untranslated region of mRNA, inducing degradation and repression of translation [37]. miRNAs also play roles in the nucleus such as regulating the stability of nuclear transcripts, chromatin remodeling at specific gene loci in order to positively or negatively regulate transcription, and alternative splicing [38]. Some long noncoding RNAs (lncRNAs) were suggested to function in numerous cellular processes such as locus-specific/genome-wide modifications of histones and chromatin remodeling, formation of nuclear subdomains, transcriptional regulation, post-transcriptional processing and transport of RNAs, shuttling between the nucleus and cytoplasm, control of translation, genomic imprinting, and X-chromosome inactivation [39]. However, the precise roles of the majority of lncRNAs remain unknown.

### 4.3. Histone Modifications

The chemical modifications of histone N-terminal tails include mono-, di-, and tri-methylation, acetylation, phosphorylation, ubiquitination, and SUMOylation [40]. These modifications of histones play an important role in gene expression or DNA replication by regulating the accessibility of DNA to the transcriptional machinery or DNA replication machinery. The modifications of histones also function as a so-called “histone code” to attract other regulatory proteins to modify chromatin structure and transcription [41,42]. Cross-talk among histone modifications to enable fine-tuning of gene expression has also been reported [43,44].

## 5. Epigenetic Regulation of ALS

### 5.1. Studies Using Human Cell and Mouse Models

#### 5.1.1. DNA Methylation

DNA methyltransferase1 (Dnmt1) is a maintenance DNA methyltransferase, whereas Dnmt3a is a de novo DNA methyltransferase. Dnmt1, Dnmt3a, and 5-methylcytosine are upregulated in the brain, and notably in the spinal cord of ALS patients [45]. Mice lacking Dnmt3a exhibited hypoactivity, defects in neuromuscular function, and decreased numbers of motor neurons, similar to the mouse ALS models targeting *SOD1* [46]. Compared with wild-type mice, the level of Dnmt3a in the mitochondria was significantly reduced in the central nervous system (CNS) and skeletal muscle of these ALS model mice [47]. Overexpression of Dnmt3a, but not Dnmt1, induced the degeneration of neurons, whereas downregulation of Dnmt3a repressed apoptosis of cultured neurons. Moreover, inhibition of Dnmt activities by specific inhibitors, RG108 and procainamide, protected motor neurons from hypermethylation of DNA and apoptosis in both cell culture and ALS model mice [13,48]. Thus, Dnmts are important in the pathogenesis of ALS (Table 2).

#### 5.1.2. Noncoding RNAs

Differential expression of various miRNAs was reported in the motor neurons of ALS patients [49,50]. For example, miR-155 and miR-142 targeting the *UBQLN2* gene and other genes associated with neurodegeneration are upregulated [51]. Moreover, genes involved in immune responses are dysregulated in ALS patients accompanying abnormal patterns of DNA methylation [49]. Other studies reported that miR-206 was upregulated in both skeletal tissues from ALS model mice targeting *SOD1* and the plasma of ALS patients [52]. In addition, other miRNAs, such as has-miR-46469-5p and has-miR4299, are upregulated and downregulated, respectively, in ALS patients [53]. Both of these miRNAs target the *EPH receptor A4* (*EPHA4*) gene that is associated with ALS in mouse and rat models [54]. Furthermore, a number of miRNAs, including miR-338 and miR-638, were reported to be dysregulated in the leukocytes of ALS patients [55]. Both miR-338 and miR-638 are involved in FTD and hereditary spastic paraplegia [56,57]. In addition, FUS is involved in the biogenesis of miRNAs, including miR-409 and miR-495 [58]. Thus, many miRNAs play crucial roles in the pathogenesis of ALS (Table 2).

Expansion of the hexanucleotide GGGGCC repeat in the *C9orf72* gene was identified as one of the causes of ALS/FTD [59,60,61,62]. The *C9orf72* gene with repeat expansion is transcribed in both the sense and antisense orientations to form nuclear and cytoplasmic sense and antisense RNA foci containing TDP-43 and FUS. This suggests that the sequestration of TDP43 and FUS in this cellular compartment plays a role in the pathogenesis of ALS/FTD associated with *C9orf72* [60,63,64]. Furthermore, TDP-43 and FUS may be involved in regulating the expression of lncRNAs such as *NEAT1* and *MALAT1*. The depletion of TDP-43 or FUS upregulated both *NEAT1* and *MALAT1* lncRNAs in mouse ALS/FTD models (Table 2) [65,66]. It was also reported that in the early stage of ALS pathogenesis, the formation of *NEAT1* RNA foci increased post-mortem as a result of the abnormal subcellular localization of TDP-43 (Table 2) [67].

#### 5.1.3. Histone Modifications

The histone acetyltransferases (HATs) CBP and p300 play important roles as coactivators of signal-dependent transcription factors. Pull-down assays with flag-tagged CBP and HeLa whole cell extracts identified FUS as a binding factor to CBP [68]. FUS also binds p300 and inhibits HAT activities of both CBP and p300 proteins [68]. Knockdown of *p300/CBP* using specific small interfering RNAs (siRNAs) reduced histone acetylation levels of lysine 9 and 14 of histone H3 (H3K9K14) in the *CyclinD1* promoter region that was assayed by chromatin immunoprecipitation, in addition to *CyclinD1* mRNA levels [68]. These observations suggest crucial roles of these coactivators in the activation of the *CyclinD1* gene. Microsatellite-based genetic association studies revealed that Elongator complex protein 3 (ELP3) harboring HAT activity is associated with motor neuron degeneration in ALS [69]. ELP3 acetylates lysine 14 of histone H3 (H3K14) and lysine 8 of histone H4 (H4K8) [70]. These studies demonstrated the link between causative genes of ALS and HAT (Table 2).

Histone deacetylases (HDACs) catalyze the removal of acetyl groups from histones and other proteins. HDACs 1-11 are Zn^2+^-dependent HDACs, whereas Sirtuins (SIRT) 1-7 are nicotine adenine dinucleotide (NAD+)-dependent deacetylases with mono-ADP-ribosyl transferase activities [71]. HDAC1, HDAC2, HDAC3, and HDAC8 are localized in the nucleus, and exhibit ubiquitous expression in multiple organs and tissues, with relatively high expression of HDAC2 and 3 in the brain [71]. HDAC2 expression is upregulated in the motor cortex and spinal cord, particularly in the nuclei of motor neurons of ALS patients [72]. A previous study revealed that HDAC1 mis-localizes to the cytoplasm in mouse ALS models targeting *FUS* [73]. Subcellular localization of HDAC1 is regulated by phosphorylation of its serine residues and the neuroprotective function of HDAC1 accumulation in the nucleus was also reported in a mouse model [74]. HDAC6 carries two catalytic domains and functions in the cytoplasm, where HDAC6 deacetylates α-tubulin to alter the stability of microtubules [75]. It was previously reported that TDP-43 and FUS interact to form a ribonucleoprotein complex that regulates the expression of HDAC6 through its mRNA stability [76]. Mutant SOD1 proteins associated with ALS are prone to aggregation. HDAC6 selectively interacted with the mutant SOD1 and knockdown of *HDAC6* increased aggregation of the mutant SOD1 in cultured human cells (Table 2) [77]. Expression of the aggregation-prone mutant SOD1 protein increased α-tubulin acetylation. Based on these observations, the mutant SOD1 associated with ALS can alter HDAC6 activity and increase α-tubulin acetylation, which consequently results in facilitation of the microtubule- and retrograde transport-dependent aggregation of mutant SOD1 [77].

Among SIRT1–7, SIRT1 associates with euchromatin and can be shuttled to the cytoplasm [78]. SIRT6 preferentially localizes in heterochromatin and SIRT7 mainly localizes in the nucleolus. SIRT2 mainly localizes in the cytoplasm, playing an important role in regulating cytoskeletal dynamics. SIRT3, SIRT4, and SIRT5 localize in mitochondria [79]. In ALS model mice carrying the *SOD1* G93A mutation, expression of both SIRT1 and SIRT2 was downregulated in neurons [80,81]. SIRT1 deacetylates and activates Peroxisome proliferator-activated receptor gamma coactivator 1α (PGC1-α), a transcriptional coactivator involved in energy production, by stimulating mitochondrial biogenesis and the respiration rate [82]. Of note, decreases in mRNA and protein levels of PGC1-α in muscle and the spinal cord were observed in both mouse models and ALS patients [83]. Thus, HDACs play important roles in the pathology of ALS (Table 2) [13,84].

Histone methylation occurs at lysine residues of histone H3 (H3K4, H3K9, H3K27, H3K36, and H3K79) and histone H4 (H4K20), and at arginine residues of histone H3 (H3R2, H3R8, H3R17, and H3R26) and H4 (H4R3) [85,86]. Lysine residues of histones can be mono-, di-, or tri-methylated, whereas arginine residues can be mono- or di-methylated. Histone methylation of lysine residues is controlled by histone lysine methyltransferases (KMTs) and demethylases (KDMs) [87,88]. Up- and downregulation of these enzymes play important roles in many pathological processes [87,88]. Methylation of arginine residues of histones is catalyzed by the protein arginine methyltransferase (PRMT) family [89]. The important role of arginine methylation in various human diseases was recently reported [90]. Dipeptide repeat expansion in the *C9orf72* gene causes a significant proportion of ALS cases [8]. Chromatin immunoprecipitation assays using antibodies against tri-methylated H3K9, H3K27, H3K79, and H4K20 revealed that the tri-methylated residues tightly bind to the expanded repeats of *C9orf72* in the brain of ALS patients [91]. *C9orf72* mRNA levels also decrease in the frontal cortices and cerebellum of ALS patients [91]. Moreover, treating fibroblasts derived from repeat carriers with 5-aza-2-deoxycytidine, a DNA and histone demethylating agent, not only reduced *C9orf72* binding to tri-methylated histone residues, but also restored *C9orf72* mRNA levels [91]. Lysine tri-methylation of histones H3 and H4 is thus involved in the reduction of *C9orf72* mRNA expression in ALS patients harboring repeat expansion in the *C9orf72* gene. Co-immunoprecipitation combined with liquid chromatography–mass spectrometry (LC-MS) analysis identified PRMT1 as a factor associated with FUS carrying R521C mutation (FUS-R521C) found in ALS patients [92]. Based on this method, the cytosolic FUS-R521C-positive stress granule aggregates contained PRMT1 [92]. Overexpression of PRMT1 rescued neurite degeneration induced by FUS-R521C under oxidative stress [92]. In contrast, loss of PRMT1 increased the accumulation of the aggregates and neurite degeneration [92]. Of note, the mRNA of *Nd1-L* encoding actin-stabilizing protein was also sequestered into the FUS-R521C/PRMT1 aggregates [92]. Overexpression of Nd1-L accordingly rescued neurite degeneration induced by FUS-R521C under oxidative stress, whereas the loss of Nd1-L promoted neurite degeneration [92]. Thus, histone methylation plays important roles in the pathogenesis of ALS (Table 2).

### 5.2. Studies Using Drosophila Models

*Drosophila* is currently emerging as a useful model for investigating human diseases, including ALS, for the purpose of finding novel diagnosis markers or targets for therapy [8]. DNA methylation levels in *Drosophila* are relatively low compared with mammals and studies on DNA methylation using *Drosophila* ALS models have not been carried out. We therefore focused on studies on noncoding RNAs and histone modifications in this section.

#### 5.2.1. Noncoding RNAs

Neuron-specific expression of human TDP-43 in *Drosophila* induces neuronal degeneration, shorter lifespan, and locomotive defects, reproducing the major symptoms of ALS/FTD. Eye imaginal disc-specific expression of human TDP-43 induces the rough eye phenotype accompanied by retinal degeneration [93,94]. Genetic modifier screening using these transgenic fly lines identified 12 out of 2933 fly lines with an exacerbated TDP-43-induced phenotype and 23 fly lines exhibiting suppression of the phenotype [95]. In this genetic screen, knockdown of the *suppressor of triplolethal* (*Su(Tpl*)) gene, also known as the *elongation factor for RNA polymerase II* (*Ell*), effectively suppressed the TDP-43-induced morphological defects of the compound eye and internal retina [95]. Furthermore, upregulation of *Su(Tpl*) enhanced the TDP-43-induced defects [95]. The protein encoded by *Su(Tpl*) is a component of two complexes: little elongation complex (LEC) and super elongation complex (SEC) [96,97,98]. In *Drosophila*, LEC contains Su(Tpl), Eaf, Ice1, and Ice2, regulating the initiation and elongation of RNA polymerase II (Pol II)-transcribed genes encoding small nuclear RNAs (snRNAs) [98,99]. SEC is composed of Su(Tpl), Eaf, Ear, Lilli, and P-TEFb [98,100]. In the ALS model flies expressing TDP-43, several noncoding RNAs were upregulated and knockdown of *Su(Tpl*) restored the levels of the increased noncoding RNAs back to normal without affecting TDP-43 expression [95]. Among these RNAs, U12 snRNA and a stress-induced *hsrω* lncRNA were found to be responsible for TDP-43-induced defects [95]. Of note, *Sat III* lncRNA, the possible human homologue of *hsrω,* is upregulated in a human cellular disease model and FTD patient tissues. The *Drosophila Su(Tpl*) gene is orthologous to the human *elongation factor for RNA polymerase II* (*ELL*) gene and *ELL2* gene. Physical interaction between TDP-43 and human ELL2 was also demonstrated by co-immunoprecipitation assay using human cells [95]. Thus, Su(Tpl)-containing complexes LEC and SEC were suggested to play important roles in TDP-43-associated toxicity.

Pan-neuron-specific knockdown of *caz* induced a short life span and locomotive defects, accompanied by short synapse length at the NMJ compared with control flies. The eye imaginal disc-specific knockdown of *caz* induced the rough eye phenotype. Pan-neuron- and motor-neuron-specific knockdown of *hsrω* lncRNA similarly induced locomotive defects and abnormal NMJ structure [101]. In the *hsrω* knockdown flies, *caz* mRNA levels were reduced and abnormal cytoplasmic localization of the caz protein was also induced. In addition to *hsrω*, knockdown of *CR18854* lncRNA suppressed the locomotive defects and short synapse branch length phenotype at the NMJ induced by the *caz* knockdown, whereas knockdown of other lncRNAs, such as *CR32207* and *CR43988*, had no effect (Table 1) [102]. Considering these observations in *Drosophila*, some human lncRNAs may similarly function in the pathogenesis of ALS/FTD associated with *FUS*.

The wild-type human FUS (*hFUSwt*) expressed in *Drosophila* is mainly soluble and neurotoxic, inducing the rough eye phenotype with the loss of pigmentation, short life span, locomotive defects, decreased numbers of synaptic bouton and active zone at the NMJ, degeneration of axons, and enlarged motoneurons [19,103,104,105,106,107]. Knockdown of *hsrω* lncRNA in the *hFUSwt*-expressing flies reduced *hFUSwt* mRNA levels and induced the formation of insoluble inclusions containing the nontoxic hFUSwt protein and LAMP1 in the cytoplasm [108]. These mechanisms may be effective to titrate soluble hFUSwt to suppress its neurotoxicity. Thus, lncRNA-dependent pathways play important roles in controlling aggregation-prone heterogeneous ribonucleoproteins (hnRNPs).

Although the *Drosophila* genome contains no orthologue of the human *C9orf72* gene, *Drosophila* models for ALS/FTD associated with *C9orf72* have been established by expressing the expanded 30 GGGGCC repeats with a CTCGAG interruption (iGGGGCC). Expression of iGGGGCC repeats in eye imaginal discs induced the rough eye phenotype in adults and that in motor neurons induced locomotive deficits [109]. The following three distinct *Drosophila* models were established to distinguish between the toxic gain-of-function mechanisms of expanded repeat RNAs and proteins produced by Repeat-associated non-AUG (RAN) translation: (1) model flies expressing expanded pure GGGGCC repeats, (2) model flies expressing the expanded GGGGCC repeat RNA interrupted with a stop codon that only produces the expanded RNAs, (3) model flies expressing non-GGGGCC RNAs with alternative codons that only produce translated proteins [24]. By analyzing these fly lines, the expanded GGGGCC repeats were suggested to induce neurotoxicity mediated by the translated proteins and RNA foci may not be a direct cause of neurodegeneration, at least in *Drosophila* models.

Previous studies revealed that overexpression of aubergine, a Piwi-family protein (PIWI) that is responsible for the biogenesis of PIWI-interacting RNA (piRNA), enhanced the phenotype of the neuron-specific *caz* knockdown flies (Table 1) [110]. Thus, various noncoding RNAs function in the pathogenesis of ALS.

#### 5.2.2. Histone Modifications

Ethyl methanesulfonate (EMS)-based mutagenesis screening was applied to identify *Drosophila* genes required for synaptic transmission and the development/survival of neurons. The screen identified loss-of-function mutations in the *Drosophila ELP3* gene [69]. Similar to human ELP3, *Drosophila* ELP3 protein exerting HAT activity is associated with the RNA polymerase II complex, and is involved in RNA processing, RNA elongation, histone acetylation, and a process related to free radical reactions. It was previously reported that ELP3 controls the expression of the gene encoding Heat shock protein 70 (HSP70) by acetylating histones H3 and H4 [111]. HSPs are well-known molecular chaperones whose expression increases in response to cellular stress. In general, motor neurons have a high threshold for activating HSPs and are therefore vulnerable to stressors such as mutant SOD1 [112]. Therefore, ELP3 was suggested to increase the expression of HSP70, thereby protecting motor neurons from degeneration. Consistent with this, intraperitoneal administration of HSP70 extended the lifespan of transgenic mice carrying the mutant *SOD1* gene [113].

*Drosophila* has five HDACs: Rpd3, HDAC3, HDAC4, HDAC6, and HDAC11. Rpd3 is a homologue of yeast Rpd3, and exhibits nearly equal homology to human HDAC1 and HDAC2. As described above, eye imaginal disc-specific expression of human TDP-43 induced the rough eye phenotype accompanied by retinal degeneration [93,94]. Knockdown of *Rpd3* effectively suppressed the rough eye phenotype induced by human TDP-43 [114]. Furthermore, TDP-43 interacts with HDAC1 via its RNA-binding domains, RRM1 and RRM2, carrying the two major TDP-43 acetylation sites, lysine 142 and 192 [114]. The 2D gel-analysis of immunoprecipitated TDP-43 from cultured human cells expressing both TDP-43 and HDAC1 revealed that the acetylation level of TDP-43 was lower than that in cells expressing TDP-43 alone [114]. Thus, HDAC1 likely plays a role in TDP-43-induced toxicity by regulating the acetylation state of TDP-43.

Bruchpilot (BRP) is a major component of the presynaptic density that tethers vesicles in *Drosophila* [115]. Immunoprecipitation assay combined with Western blotting revealed that acetylated lysine residues of BRP increased in *HDAC6* null mutants and decreased in flies overexpressing *HDAC6* [111]. In addition, in vitro assay revealed that HDAC6 deacetylates BRP [116]. Thus, HDAC6 is both necessary and sufficient for BRP deacetylation [116]. In mutant flies for *TBPH*, a *Drosophila* homologue of *TDP-43*, *HDAC6* mRNA levels were reduced, whereas in flies expressing human wild-type TDP-43 or TDP-43 carrying pathogenic mutations, *HDAC6* mRNA was upregulated [116,117]. Decreased BRP acetylation, larger vesicle-tethering sites, and increased neurotransmission were also observed in flies expressing the mutant form of TDP-43 [116]. Of note, these defects were similar to those observed with the expression of *HDAC6* and opposite to *HDAC6* null mutants. Consequently, reduced levels of HDAC6 or increased levels of ELP3, which acetylates BRP, rescued the presynaptic density deficits in TDP-43-expressing flies and the locomotive defects in adults [116]. These studies suggest a link between HDAC6 and ALS pathogenesis.

#### 5.2.3. Genetic Screening of Epigenetic Regulators Using the Drosophila Model Targeting Caz

Knockdown of *caz* induced the rough eye phenotype as described above [18]. The rough eye phenotype is a convenient visible marker for large-scale genetic screens, enabling the identification of genes that genetically interact with the gene of interest [18,118,119]. The identified genes by this rough eye modifier screen are critical because they normally act as rate-limiting factors in the cellular pathways related to the gene of interest. In *Drosophila*, there are 91 epigenetic regulators that are classified by Gene Ontology. Mutants or RNAi lines for these epigenetic regulator genes were collected from the *Drosophila* stock centers and crossed with the eye imaginal disc-specific *caz* knockdown flies, and adult compound eyes of the progenies were examined by scanning electron microscopy (SEM). By this genetic screening, we identified 14 genes whose mutation or knockdown effectively suppressed the rough eye phenotype induced by *caz* knockdown, whereas there were 5 genes whose mutation or knockdown strongly enhanced the rough eye phenotype (to be published elsewhere). The 14 suppressor genes included *Histone acetyltransferase 1 (Hat1)* [120,121], *chameau* (*chm*) [122,123], *N(alpha)-acetyltransferase 60 (Naa60)* [124], *Negative Cofactor 2β (Nc2β)* [125], *Spt7* [126], *TBP-associated factor 1(Taf1)* [127], *will die slowly (wds)* [125], *Sirtuin 2 (Sirt2)* [128], *Methyltransferase 2(Mt2)* [129], *extra sexcombs (esc)* [130], *grappa (gpp)* [131], *SET domain containing 1 (Set1)* [132], *Tousled-like kinase (Tlk)* [133], and *Vacuolar protein sorting 11(Vps11)* [134]. Among them, mutations of *chm* and *Naa60* suppressed the locomotive defects and morphological defects of synapse at the NMJ induced by *caz* knockdown (to be published elsewhere). Further analyses are necessary to understand the biological meaning of the possible link between the identified genes and *caz*. The five enhancer genes identified by the rough eye modifier screen include *enoki mushroom (enok)* [135], *Nucleolar protein 66* (*NO66*) [136], *eggless (egg)* [137], *PR/SET domain containing protein 7 (PR-Set7)* [138], and *ballchen (ball)* [139]. Further investigation of these modifier genes in relation to *caz* and other causative genes of ALS is of interest.

## 6. Epigenetic Regulation of CMT

### 6.1. Studies Using Human Cell and Mouse Models

Studies of epigenetic mechanisms involved in CMT are in the initial stage. In particular, studies on DNA methylation in relation to CMT have not been reported; therefore, we focused on noncoding RNAs and histone modifications in this section.

#### 6.1.1. Noncoding RNAs

Although CMT type 1A (CMT1A) is known to be caused by increased gene dose of *PMP22*, the age of onset and severity of symptoms vary considerably among individual patients [25]. A common single nucleotide polymorphism was found in the miRNA miR-149, which was predicted to target several CMT-inducing genes, including *PMP22* [140]. The polymorphism was located around the 3′ end of the precursor RNA of miR-149. Association studies between the polymorphism and clinical phenotypes of CMT1A patients revealed that the polymorphism in miR-149 was closely related to the onset age and severity of symptoms, suggesting that the variant in miR-149 is a genetic modifier that can cause the phenotypic heterogeneity of CMT1A (Table 2) [140].

#### 6.1.2. Histone Modifications

HDAC6 functions as the major deacetylating enzyme of α-tubulin and is associated with several neuropathies, including ALS, as described above [75]. Mutations in the *27-kDa small heat-shock protein* (*HSPB1*) gene can cause CMT type 2F [141]. CMT model mice carrying two different *HSPB1* mutations (S135F and P182L) exhibited characteristics of CMT depending on the mutation [142]. It was also reported that expression of mutant HSPB1 reduced acetylated α-tubulin levels and induced defects in axonal transport. Inhibition of HDAC6 increased acetylated α-tubulin levels and simultaneously rescued the axonal transport defects together with the CMT-like phenotype of the model mice. Mutations in the *glycyl-tRNA synthetase* (*GARS*) gene have been identified in CMT-type 2D patients [143]. In CMT model mice carrying C201R mutant GARS, peripheral nerves and dorsal root ganglia exhibited reduced levels of α-tubulin acetylation accompanied by short neurite length and defects in mitochondrial transport [144]. Physical interaction between the glycyl-tRNA synthetase and HDAC6 was also demonstrated by co-immunoprecipitation, and this interaction was inhibited by the HDAC6 inhibitor tubastatin A [144]. The inhibition of HDAC6 also rescued defects in transport of mitochondria in neurons of the model mice. Thus, HDAC6 plays a critical role in the pathogenesis of CMT (Table 2).

The presence of mono-ubiquitination of histone H2B at lysine 120 (H2Bub1) located immediately downstream of the transcription initiation sites of genes may be responsible for active gene transcription by promoting di-and tri-methylation of lysine 4 and lysine 79 of histone H3, thereby increasing the processivity of RNA polymerase II with the recruitment of transcription elongation machinery [145,146]. In mice with conditional depletion of the subunit of E3 ligase responsible for mono-ubiquitination, myelination in Schwann cells abnormally produced thin unstable myelin, which resulted in a peripheral neuropathy characterized by hypomyelination and progressive degeneration of axons similar to the features of CMT [147]. H2B mono-ubiquitination induces the high expression of myelin and lipid biosynthesis genes while repressing immaturity genes. Recruitment of the E3 ligase by Early growth response 2 (Egr2), a key transcription factor for peripheral myelination, to its target genes is also known to be required for this process (Table 2) [147].

Mutations in the ATPase domain of Microchidia CW-type zinc finger 2 (MORC2) have recently been implicated in axonal CMT [148]. Genome-wide genetic screening identified the *MORC2* gene as an important one for epigenetic silencing by the Human Silencing Hub (HUSH) complex. Of note, HUSH recruits MORC2 to target sites in heterochromatin [149]. Depletion of MORC2 induced chromatin decompaction at these target sites concomitant with the reduction of tri-methylation of H3K9 and transcriptional de-repression. The ATPase activity of MORC2 is required for HUSH-mediated silencing. The most common mutation found in CMT patients (R252W) is in the ATPase domain, which hyper-activates HUSH-mediated silencing in neuronal cells, suggesting an underlying mechanism in the role of *MORC2* mutations in the pathogenesis of CMT [150].

### 6.2. Studies Using Drosophila Models

#### 6.2.1. Noncoding RNAs

Rough eye modifier screening using the CMT model flies targeting *dFIG4* was carried out to identify genetic interactants with *dFIG4*. By this screen, the *CR18854* gene encoding a lncRNA consisting of 2566 bases was identified as a gene whose depletion effectively suppressed the rough eye phenotype induced by *dFIG4* knockdown [102]. Mutation and knockdown of *CR18854* also suppressed the enlarged lysosome phenotype induced by Fat body-specific *dFIG4* knockdown [102]. Of note, *CR18854* genetically interacted with not only *dFIG4*, but also *caz*, a causative gene of ALS, whereas *hsrω* lncRNA was associated with ALS but genetically interacted with *dFIG4* [102]. Both of these lncRNAs were therefore suggested to be involved in a common pathway between CMT and ALS pathogenesis [102]. More recently, another lncRNA, *CR43467*, was identified as a genetic interactant with *dFIG4* [151], whereas the depletion of other lncRNAs, such as *CR32207* and *CR43988,* exerted no apparent effect on the phenotype induced by *dFIG4* knockdown [102]. Knockdown of *CR43467* rescued the aberrant morphology of synapse at the NMJ and the vacuole enlargement phenotype induced by *dFIG4* knockdown. This lncRNA genetically interacted with other genes associated with peripheral neuropathy [151], such as *cytochrome C oxydase assembly factor 7* (*COA7*) [152], *hydroxyacyl-CoA dehydrogenase/3-ketoacyl-CoA thiolase/enoyl-CoA hydratase beta* (*HADHB*) [153], and *pyruvate dehydrogenase beta* (*PDHB*), in *Drosophila* [154]. COA7 controls the assembly of the mitochondrial respiratory chain complexes responsible for oxidative phosphorylation and its depletion causes spinocerebellar ataxia with axonal neuropathy type 3 [152]. HADHB is a subunit of the mitochondrial trifunctional protein (MTP) that is required for the beta-oxidation of fatty acids to produce acetyl-CoA [153]. Mutations in the *HADHB* gene induce defects in beta-oxidation that result in MTP deficiency. MTP deficiency is associated with cardiomyopathy, recurrent Leigh-like encephalopathy, the later onset of peripheral neuropathy, and myopathy with or without episodic myoglobinuria [153]. PDHB is a subunit of pyruvate dehydrogenase (PDC) that is involved in the production of acetyl-CoA [154]. Deficiency of PDC causes deficits in mitochondrial function and is associated with peripheral neuropathies [154]. The nucleotide identity between *CR43467* and *CR18854* was 34.9%, being relatively high, suggesting a similar cellular function for both lncRNAs [151]. Based on the nucleotide sequence homology, there may be no human homologue of *CR43467* and *CR18854*, although it does not exclude the possibility of functional homologues. As the knockdown of *CR43467* or *CR18854* effectively rescued the deficits induced by the *dFIG4* knockdown in *Drosophila*, their functional human homologues, if present, may be promising targets for CMT therapies.

#### 6.2.2. Histone Modifications

The human SLC25A46 protein is involved in mitochondrial dynamics by regulating the association of mitochondria with the endoplasmic reticulum and/or by controlling the oligomerization of MFN1 and MFN2 [155,156]. Mutations in the *SLC25A46* gene are associated with the so-called mitochondrial diseases that are sometimes classified as CMT type 2, optic atrophy, and Leigh syndrome [157,158]. The diverse symptoms of diseases caused by mutations in *SLC25A46* may be related to the dysregulation of epigenetic mechanisms. There are two homologues of human *SLC25A46* in *Drosophila,* designated as *dSLC25A46a* (*CG8931*) [159] and *dSLC25A46b* (*CG5755*) [160]. *Drosophila* models targeting *dSLC25A46a* or *dSLC25A46b* exhibited similar phenotypes such as locomotive defects and aberrant morphology of synapses at the NMJ [159,160]. Gene Expression Omnibus (GEO) analysis revealed that HDAC1 is associated with several *SLC25A46* genomic regions in human cultured CD4-positive cells, and this association was further evaluated using the *Drosophila* models [161]. Rpd3, a *Drosophila* HDAC1/HDAC2 homologue, was found to regulate the histone H4K8 acetylation state in the *dSLC25A46b* genomic regions [161]. Depletion of *Rpd3* effectively rescued the locomotive defects and morphological deficits of synapses at the NMJ induced by *dSLC25A46b* knockdown [161]. These studies using *Drosophila* models suggest that HDAC1 is involved in the pathogenic process of CMT.

## 7. Perspectives

Genetic screening using *Drosophila* models identified several lncRNAs as genetic interactants with causative genes for ALS and CMT. Although exome analyses are usually performed for the genetic diagnosis of ALS and CMT, these findings in *Drosophila* models suggest that the information on nonprotein coding regions in the human genome is also important to understand the pathogenic processes of both ALS and CMT. In both mammals and *Drosophila*, the in vivo roles for the majority of lncRNAs remain unknown. Although *Drosophila* is currently drawing attention as a useful model for investigating both ALS and CMT, bioresources for lncRNAs in *Drosophila*, such as mutants or knockdown strains, remain limited. Therefore, for the characterization of lncRNAs in relation to the pathogenesis of ALS and CMT, development of bioresources in *Drosophila*, such as a collection of knockdown lines for lncRNAs created by the CRISPR-dCas9 method [151], may be necessary. Further studies on *Drosophila* lncRNAs in relation to the causative genes for ALS and CMT in humans will expand our knowledge of human lncRNAs, and aid in finding the novel gene diagnosis markers and therapeutic targets for ALS and CMT.

As described above, genetic screening using *Drosophila* models identified several epigenetic regulators as genetic interactants with causative genes for ALS and CMT. Although the analyses are still preliminary in some cases, identification of these genetic interactants may aid in future studies. For example, *Drosophila* provides a useful tool for clarifying how environmental stress affects epigenetic regulators. Indeed, Rpd3 was reported to be responsible for starvation stress tolerance [162]. Nuclear accumulation of Rpd3 occurs in the early stage of starvation, resulting in the upregulation of rRNA synthesis and increase in stress tolerance proteins to acquire starvation stress tolerance [162]. Furthermore, the *Drosophila* histone methyltransferase G9a (dG9a) plays a crucial role in acquiring tolerance to starvation stress by maintaining energy storage, such as amino acid, trehalose, glycogen, and triacylglycerol levels, during starvation [163]. Regarding starvation-induced behavioral changes, starved flies increase their locomotion activity to increase the probability of finding desirable foods [88,164]. dG9a functions as a key regulator of the decision of behavioral strategy under starvation conditions [164]. Studies of CMT using *Drosophila* models only started 10 years ago, and studies of epigenetic mechanisms involved in ALS and CMT are still in the initial stage. Characterization of epigenetic regulators in relation to ALS and CMT is one future direction in this academic field, and *Drosophila* models will provide a powerful genetic tool for this purpose.

The *Drosophila* models have been used as powerful tools for screening of various drugs [165]. *Drosophila* has an open blood vascular system that enables drugs to be delivered to any target organ, including the brain [166]. Although the scale of drug screening with *Drosophila* models is smaller than the high-throughput screening of a library of small compounds in vitro or in cultured cells, in vivo screening with *Drosophila* models is expected to provide a high-quality hit [166]. Therefore, screening of Food and Drug Administration-approved drugs and natural substances to develop a potential therapy for ALS and CMT would be more extensively performed using *Drosophila* models. The pioneer studies reported that honeybee products (coffee honey from *Apis cerana*) improved locomotive and learning abilities of the *Drosophila* ALS/FTD model targeting *Ubiquilin* [167].

## Figures and Tables

**Table 1 ijms-22-00491-t001:** Epigenetic regulators that affected the rough eye and/or neural phenotype induced by *caz* knockdown.

Gene	Function	Transcripts	Alleles	Human Orthologue	Effect on Rough Eye and Neural Phenotype	References
heat shock RNA ω (hsrω)	lncRNA, nuclear speckle organization, positive regulation of apoptotic process, protein metabolic process	7	30	none	suppression	[102], https://flybase.org/reports/FBgn0001234
CR18854	lncRNA, hairpin RNA	3	5	none	suppression	[102], https://flybase.org/reports/FBgn0285991
CR32207	lncRNA, hairpin RNA	1	2		no effect	[102], https://flybase.org/reports/FBgn0052207
CR43988	lncRNA	1	1		no effect	[102], https://flybase.org/reports/FBgn0264720
aubergine (aub)	binding and biogenesis of piRNA, gamete generation, gene silencing, mRNA poly(A) tail shortening	2	50	PIWIL1, PIWIL2, PIWIL3, PIWIL4	enhancement	[110], https://flybase.org/reports/FBgn0000146

**Table 2 ijms-22-00491-t002:** Summary of studies with human cells and mouse models.

	Epigenetic Mechanism	Related Epigenetic Factor	References
ALS/FTD	DNA methylation	Dnmt1, Dnmt3	[13,45,46,47,48]
	noncoding RNA	miR-155, miR-142, miR-206, has-miR-46469-5p, has-miR4299, miR-338, miR-638, miR-409, miR-495	[51,52,53,55,56,57,58]
		NEAT1, MALAT1	[65,66,67]
	histone modification	CBP, p300, ELP3	[68,69,70]
		HDAC1, HDAC2, HDAC6, SIRT1, SIRT2	[72,73,74,75,76,77,79,80,81,82]
		PRMT1	[92]
CMT	noncoding RNA	miR-149	[140]
	histone modification	HDAC6	[144]
		E3 ubiquitin ligase, Egr2	[147]

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
