# Peer review of "Epigenetic Regulation of ALS and CMT: A Lesson from Drosophila Models"

_ijms, 2021, doi:10.3390/ijms22020491_

Round 1

Reviewer 1 Report

The authors perfectly and extensively illustrated the usefulness of Drosophyla to study epigenetic regulation of Amyotrophic Lateral Sclerosis and Charcot Marie Tooth diseases.

This manuscript is a model of the State of the Art and should be acknowledge for this comprehensive review of the topic referred by 165 appropriate references. I am convinced it will be a highly respected and cited manuscript.

Author Response

The authors perfectly and extensively illustrated the usefulness of Drosophila to study epigenetic regulation of Amyotrophic Lateral Sclerosis and Charcot Marie Tooth diseases.

This manuscript is a model of the State of the Art and should be acknowledge for this comprehensive review of the topic referred by 165 appropriate references. I am convinced it will be a highly respected and cited manuscript.

     Thank you very much for your supportive comments.

Reviewer 2 Report

The manuscript suggests a comprehensive overview of the contribution of epigenetic regulation for two important diseases, Amyotrophic lateral sclerosis (ALS) and Charcot-Marie-Tooth disease (CMT).

Interestingly, the authors focus on Drosophila as emergent model for both diseases.

This study opens up new interesting perspectives into understanding the link between these two pathologies.

There are some minor comments in the manuscripts which need to be addressed.

  • The authors should indicate the selection criteria on which the manuscript writing was based, such as which types of database sources were used (e.g. Scopus, MEDLINE, PubMed, ScienceDirect, etc.), how the filter occurred (types of studies considered and time window analysed), which critically justify the manuscript construction.
  • The authors should briefly describe the similarities and differences that occur when comparing both diseases, especially in relation to pathological mechanisms and aetiological factors. I would suggest introducing a short, concise paragraph on this before delving into the main topic of epigenetic regulation.
  • are the references missing in table 1?
  • To make the reading more fluent, the authors could outline the studies on human cells and animal models, for both ALS and CMT, in a table.

Author Response

The manuscript suggests a comprehensive overview of the contribution of epigenetic regulation for two important diseases, Amyotrophic lateral sclerosis (ALS) and Charcot-Marie-Tooth disease (CMT).

Interestingly, the authors focus on Drosophila as emergent model for both diseases.

This study opens up new interesting perspectives into understanding the link between these two pathologies.

There are some minor comments in the manuscripts which need to be addressed.

  • The authors should indicate the selection criteria on which the manuscript writing was based, such as which types of database sources were used (e.g. Scopus, MEDLINE, PubMed, ScienceDirect, etc.), how the filter occurred (types of studies considered and time window analysed), which critically justify the manuscript construction.

     As suggested by the reviewer, we added the following sentence in Introduction “In writing this review, we searched the references by PubMed using the keywords described above” (line 58).

  • The authors should briefly describe the similarities and differences that occur when comparing both diseases, especially in relation to pathological mechanisms and aetiological factors. I would suggest introducing a short, concise paragraph on this before delving into the main topic of epigenetic regulation.

     In order to stand out similarity and difference between ALS and CMT as suggested, we added several words and sentences in Introduction as follows: “Rapid progress of …” (line 27); “The average prevalence of CMT is approximately 1: 2,500 persons, while that of ALS is about 1: 25,000 persons all over the world” (line 33); “Both ALS and CMT induce neurodegeneration” (line 52). Although the reviewer suggested to make a new paragraph for this, we are afraid that it may be a bit too redundant.

  • are the references missing in table 1?

     As suggested by the reviewer 3, we deleted unpublished data without references in the original Table 1 to make a new Table 2 (line 287).

  • To make the reading more fluent, the authors could outline the studies on human cells and animal models, for both ALS and CMT, in a table.

     As suggested by the reviewer, we made a new Table 1 to summarize studies with human cells and mouse models (line 151).

Reviewer 3 Report

The authors present a very detailed review of epigenetic ALS/CMT disease modeling in Drosophila. While there is so much detail as to be overwhelming to a casual reader, overall this review will be a valuable resource to those seeking a comprehensive survey of the field. I have several minor comments as well as some major structural suggestions. 

Minor comments:

1. Please consider changing the word "essential," which the authors use often throughout the manuscript. This word has specific meaning in the genetics field ("necessary for viability"), but the authors seem to be using its English definition ("necessary"). This is misleading. 

2. Please cite primary literature as well as reviews, especially when describing Drosophila models. If this is to be a resource for the field, it is important to cite source articles. For example, reference 8 is a review article, but it would be more helpful to cite primary literature in section 2. 

3. Table 2 does not add any information that is not in the text and seems unnecessary unless more useful information is included in the table.

Major structural suggestions:

1. I am confused by the addition of the caz knockdown experiment (including Table 1 and Figure 1, beginning at line 346). Although it would be appropriate to discuss the results of this work if it were published elsewhere, it is not yet published and is out of place and inappropriate in this review article. The authors could consider submitting the kd screen to G3 or another venue that publishes screen results, or bioRxiv so that they can cite this work in this manuscript. As is, not enough data are given to evaluate this experiment (for example, no efficacy of candidate knock down by RT-qPCR or western). Figure 1 should be presented with genotypes overlaid on the images and enhancers/suppressors of the phenotype on different rows or somehow separated. Lines 384-482 are simply a discussion of each candidate from this screen, which is not very interesting. But more importantly, these candidates have not been validated and their inclusion in a review article will make them seem as if they have. I do not believe that these screen results belong in this review manuscript. 

2. The authors separate the diseases from the Drosophila modeling, which makes it difficult to compare. For example, I suggest beginning with ALS symptoms/etiology and then discussing Drosophila models. Then CMT and Drosophila CMT models (merge and rearrange sections 1-3). Similarly, in section 5, the authors discuss epigenetic regulation of ALS (DNA methylation, then noncoding RNAs, then histone modifications) and THEN go through each of these sections again discussing Drosophila models, when it would be easier to read and compare if the authors discussed noncoding RNA studies in human cells/other organisms AND Drosophila all together. The same goes for the CMT discussion (Section 6), where I believe the discussion of disease modeling in non-Drosophila and Drosophila systems should be side-by-side.

3. I suggest the authors expand on the intriguing suggestion on line 618: that Drosophila models of ALS/CMT will be useful for screening FDA-approved drugs for therapeutic candidates. This is exciting but the manuscript ends abruptly on this note with little explanation. 

Author Response

The authors present a very detailed review of epigenetic ALS/CMT disease modeling in Drosophila. While there is so much detail as to be overwhelming to a casual reader, overall this review will be a valuable resource to those seeking a comprehensive survey of the field. I have several minor comments as well as some major structural suggestions. 

Minor comments:

  1. Please consider changing the word "essential," which the authors use often throughout the manuscript. This word has specific meaning in the genetics field ("necessary for viability"), but the authors seem to be using its English definition ("necessary"). This is misleading. 

     As suggested by the reviewer “essential” was replaced by other words such as “important” (line 64, 131, 224, 275, 418), “crucial” (line 165, 184, 497) and “critical” (line 355, 403).

  1. Please cite primary literature as well as reviews, especially when describing Drosophila models. If this is to be a resource for the field, it is important to cite source articles. For example, reference 8 is a review article, but it would be more helpful to cite primary literature in section 2. 

     As suggested by the reviewer, primary literatures (new references 18, 19, 20, 21, 22, 23 and 24) were given in the section 2 (line 65). Accordingly, other references were renumbered.

  1. Table 2 does not add any information that is not in the text and seems unnecessary unless more useful information is included in the table.

     As suggested by the reviewer, we deleted the original Table 2 from the manuscript.

Major structural suggestions:

  1. I am confused by the addition of the caz knockdown experiment (including Table 1 and Figure 1, beginning at line 346). Although it would be appropriate to discuss the results of this work if it were published elsewhere, it is not yet published and is out of place and inappropriate in this review article. The authors could consider submitting the kd screen to G3 or another venue that publishes screen results, or bioRxiv so that they can cite this work in this manuscript. As is, not enough data are given to evaluate this experiment (for example, no efficacy of candidate knock down by RT-qPCR or western). Figure 1 should be presented with genotypes overlaid on the images and enhancers/suppressors of the phenotype on different rows or somehow separated. Lines 384-482 are simply a discussion of each candidate from this screen, which is not very interesting. But more importantly, these candidates have not been validated and their inclusion in a review article will make them seem as if they have. I do not believe that these screen results belong in this review manuscript. 

     We understand the reviewer’s concern. As suggested by the reviewer, we deleted the original Figure 1 and the unpublished parts of the original Table 1 to make the new Table 2 in which some more information was added (line 287). The original paragraphs from line 384 to line 482 were also deleted from the manuscript as suggested.

  1. The authors separate the diseases from the Drosophila modeling, which makes it difficult to compare. For example, I suggest beginning with ALS symptoms/etiology and then discussing Drosophila models. Then CMT and Drosophila CMT models (merge and rearrange sections 1-3). Similarly, in section 5, the authors discuss epigenetic regulation of ALS (DNA methylation, then noncoding RNAs, then histone modifications) and THEN go through each of these sections again discussing Drosophila models, when it would be easier to read and compare if the authors discussed noncoding RNA studies in human cells/other organisms AND Drosophila all together. The same goes for the CMT discussion (Section 6), where I believe the discussion of disease modeling in non-Drosophila and Drosophila systems should be side-by-side.

     We understand the reviewer’s opinion. However, we still believe that describing non-Drosophila and Drosophila systems in the separate sections would be rather easier for the readers to follow. We therefore did not combine them in the revised manuscript. We really hope that the reviewer agrees to this decision.

  1. I suggest the authors expand on the intriguing suggestion on line 618: that Drosophila models of ALS/CMT will be useful for screening FDA-approved drugs for therapeutic candidates. This is exciting but the manuscript ends abruptly on this note with little explanation. 

     As suggested by the reviewer, we added some more explanations for the drug screening with Drosophila models by adding two new references (new references 170 and 171) (line 506-510) as follows: “The Drosophila models have been used as powerful tools for screening of various drugs (170). Drosophila has an open blood vascular system that enables drugs to be delivered to any target organ, including the brain (171). Although the scale of drug screening with Drosophila models is smaller than the high-throughput screening of a library of small compounds in vitro or in cultured cells, in vivo screening with Drosophila models is expected to provide a high-quality hit (171)”.